# Hypothalamic-Pituitary-Ovarian Axis Disorders Impacting Female Fertility

**DOI:** 10.3390/biomedicines7010005

**Published:** 2019-01-04

**Authors:** Sasha Mikhael, Advaita Punjala-Patel, Larisa Gavrilova-Jordan

**Affiliations:** 1Division of Reproductive Endocrinology Infertility and Genetics, Augusta University, Medical College of Georgia, Augusta, GA 30912, USA; lgavrilovajordan@augusta.edu; 2Department of Obstetrics and Gynecology, Augusta University, Medical College of Georgia, Augusta, GA 30912, USA; apunjala@augusta.edu

**Keywords:** ovulatory dysfunction, infertility, hypothalamic-pituitary-ovarian axis, WHO ovulation disorders, primary ovarian insufficiency

## Abstract

The hypothalamic-pituitary-ovarian (HPO) axis is a tightly regulated system controlling female reproduction. HPO axis dysfunction leading to ovulation disorders can be classified into three categories defined by the World Health Organization (WHO). Group I ovulation disorders involve hypothalamic failure characterized as hypogonadotropic hypogonadism. Group II disorders display a eugonadal state commonly associated with a wide range of endocrinopathies. Finally, group III constitutes hypergonadotropic hypogonadism secondary to depleted ovarian function. Optimal evaluation and management of these disorders is based on a careful analysis tailored to each patient. This article reviews ovulation disorders based on pathophysiologic mechanisms, evaluation principles, and currently available management options.

## 1. Introduction

The hypothalamic-pituitary-ovarian (HPO) axis must be viewed as an entity that works in concert to allow for procreation by means of cyclic production of gonadotropic and steroid hormones. This cycle is tightly regulated to select a dominant follicle for ovulation, meanwhile priming the endometrium for implantation. The ovary plays a pivotal role in the production of steroid hormones necessary for follicular development and oocyte maturation. It contains a finite number of oocytes that a woman will have for the span of her reproductive life and influences the hormonal milieu required for oocyte maturation and fertilization. This complex regulation can be negatively impacted when pathologies occur within any juncture of the HPO axis.

Ovulatory disorders are the leading cause of infertility classified into three categories according to the World Health Organization (WHO [1]). Group I ovulation disorders encompasses hypothalamic failure leading to hypogonadotropic hypogonadism and accounts for approximately 10% of ovulation disorders and includes hypogonadotropic hypogonadism, panhypopituitarism, autoimmune or infectious hypophysitis, pituitary adenomas and histiocytosis. Group II disorders involves dysfunction of the HPO axis which constitutes 85% of ovulations disorders caused by PCOS, abnormal body mass index (BMI) and endocrinopathies. Finally, group III constitutes ovarian insufficiency, previously known as ovarian failure, which has significant implications leading to oocyte depletion. Group III disorders involves multiple complex etiologies causing premature ovarian insufficiency (or failure), including genetic, iatrogenic and acquired causes.

Here, we provide an overview of ovulation disorders categorized by pathophysiologic mechanism from recent literature, along with guidance on evaluation and currently available management options.

## 2. Hypothalamic Pituitary Failure 

Group I ovulation disorders, hypothalamic-pituitary failure (HPF) manifests as hypogonadotropic hypogonadism. Females with HPF typically present with delayed or impaired pubertal development, primary or secondary amenorrhea, and infertility. Idiopathic hypogonadotropic hypogonadism (IHH) due to a congenital absence of gonadotropic releasing hormone (GnRH) is the most common cause of HPF. IHH associated with anosmia is termed Kallmann syndrome, which results from failure of GnRH neuronal migration from the olfactory placode during embryonic development. This disorder has a multimodal genetic inheritance pattern and can be transmitted in an X- linked, autosomal dominant, autosomal recessive, or oligogenic fashion with variable penetrance. Several gene mutations in association with this disorder have been identified, most notably ANOS1 [2]. The nasal embryonic LHRH factor (NeLF) mutation is one of the most recently described mutations [3]. Newer studies investigating the pathogenesis of these diseases are bridging gaps in understanding of the genetic basis of IHH.

Acquired causes include panhypopituitarism, characterized by >50% deficiency of one or more tropic hormones secreted by the pituitary gland. Etiologies leading to panhypopituitarism involve tissue necrosis (secondary to acute ischemia or pituitary gland apoplexy), autoimmune or infectious hypophysitis and compression by pituitary adenomas or adjacent brain tumors. Postpartum hemorrhage and significant head trauma may temporarily or permanently impact hormone production from the anterior pituitary. Similarly, intracranial tumors may compress the anterior pituitary stalk disrupting axonal communication between the hypothalamus and pituitary gland. Craniopharyngioma, a benign brain tumor deriving from a Rathke’s pouch, is the most common brain tumor causing delayed puberty and primary amenorrhea. In addition to dysregulation of the HPO axis, bitemporal hemianopsia and chronic headaches may be observed if the optic nerve is compressed [4]. Additionally, HPF is a common result of brain irradiation. 

Langerhans cell histiocytosis should be considered when more common causes of HPF are ruled out. This condition is marked by uncontrolled proliferation of dendritic cells associated with the monocyte-macrophage system [5]. When Langerhans cells infiltrate the anterior pituitary, disordered hormone production may lead to hypogonadotropic hypogonadism [6,7]. Furthermore, IHH may be due to septo-opto dysplasia (De Morsier syndrome), a congenital disorder characterized by hypoplasia of the optic nerve and pituitary gland. These patients additionally display craniofacial abnormalities thought to occur secondary to vascular insult in the 6th-7th week of embryogenesis [8]. 

### Evaluation and Management of Hypothalamic-Pituitary Failure

A patient’s medical history can help delineate between the various etiologies, to tailor evaluation and management. Basic evaluation includes follicle stimulating hormone (FSH), luteinizing hormone (LH), and estradiol (E2). In some instances, further evaluation is advisable to exclude central hypothyroidism (low TSH), hypocortisolism (low ACTH) and low growth hormone, which are a hallmark of panhypopituitarism.

Commonly, a head MRI is indicated to rule out morphologic pituitary anomalies or presence of a tumor [9]. Overall, management principle is based on patient needs. Women desiring conception, will benefit from ovulation induction with exogenous gonadotropins. Recombinant FSH, purified human menopausal gonadotropin (hMG) in addition to human chorionic gonadotropin (hCG) are commonly used [9,10]. Outside of procreative management, replacement of sex steroid hormones with estrogen and progesterone is required to reduce morbidity associated with a hypoestrogenic state. In particular circumstances of prepubertal females, estrogen replacement for one to two years is critical to mimic natural thelarche followed by subsequent addition of progesterone, once advanced tanner stage is achieved. These patients require careful monitoring during ovarian stimulation.

## 3. Hypothalamic-Pituitary-Ovarian Axis Dysfunction

Group II, eugonadal ovulatory dysfunction, accounts for the majority of ovulation disorders and comprises a wide spectrum of disorders. 

### 3.1. Polycystic Ovary Syndrome

Polycystic ovary syndrome (PCOS) is the most common endocrine disorder in reproductive aged women with a prevalence of 6–10% [11]. It is the most common cause of oligo-or-anovulation. Several mechanisms describing pathogenesis of PCOS includes loss of GnRH pulsatility with increased LH secretion by the pituitary gland, hyperinsulinemia, ovarian insulin resistance, theca cell dysfunction and hyperandrogenism [12]. Clinical presentation of PCOS is highly variable with four clinical phenotypes described [10]. Phenotype A is manifested as hyperandrogenism (clinical or biochemical confirmation), phenotype B is described as hyperandrogenism concurrent with ovulatory dysfunction, phenotype C comprises hyperandrogenism with polycystic ovary morphology (PCOM) without the presence of ovulatory dysfunction, and phenotype D involves PCOM with ovulatory dysfunction without signs of hyperandrogenism [11,13]. Phenotypes A and B are more commonly seen in the clinical setting because of severe metabolic abnormalities while the milder phenotype D with fewer symptoms of metabolic disturbances is more prevalent in the general population. The variation in observed phenotype is thought to be influenced by DNA methylation with presence of histones, microRNAs as well as other gene regulatory proteins. The underlying pathophysiology is multifactorial with complex polygenic inheritance [13,14]. Multiple loci have been identified by genome-wide association studies and epigenetic regulation has been explored. In one study by Chen and colleagues, miR-93, which is responsible for downregulation of GLUT4, appeared to be overexpressed in PCOS patients. GLUT4 is a protein responsible for insulin mediated glucose translocation into adipocytes and is required for glucose metabolism. Its downregulation results in insulin resistance as observed in PCOS [14]. Though several candidate genes have been found to be causative, routine genetic testing is currently not recommended. 

### 3.2. Weight

Extremes of weight and body fat distribution, influences the regulation of ovulation. Obesity, particularly central type, interferes with endocrine and paracrine mechanisms involved in reproductive cycle regulation. Loss of normal GnRH pulsatility is commonly due to excessive aromatization of androgen precursors, DHEA and testosterone, to estrone in adipose tissue, decreased levels of sex hormone binding globulin (SHBG) and elevated production leptin by adipocytes [15]. Furthermore, Gesink et al. demonstrated that women with a BMI > 29 kg/m exhibit lower fecundability as a result of suboptimal folliculogenesis adversely affecting oocyte quality [2].

In contrary, in underweight patients, GnRH pulsatility is affected by depletion of circulating leptin, excessive cortisol and neuropeptide Y production, in addition to elevated centrogenic opioid and endorphin secretion. 

### 3.3. Endocrinopathies

Hyperprolactinemia is a common cause for ovulatory dysfunction. There are many known causes of hyperprolactinemia including anterior pituitary adenomas, prolactinomas, primary hypothyroidism with thyrotropin releasing hormone (TRH) stimulation of prolactin, renal failure with abnormal clearance of prolactin, and psychotropic medications altering dopamine release. 

Primary hypothyroidism hinders regulation of the HPO axis since excessive TRH secretion will interfere with GnRH pulsatility. Furthermore, abnormal thyroid function affects folliculogenesis and oocyte quality. Some studies have suggested thyroid stimulation hormone (TSH) > 2.5 μIU/mL may be enough to lead to disordered ovulation; however additional studies have presented conflicting data. The literature has demonstrated that during controlled ovarian hyperstimulation, TSH is noted to increase and if baseline TSH exceeds 3 μIU/mL, patients will experience clinical hypothyroidism [16,17,18]. Additionally, it has been demonstrated that assisted reproductive technologies (ART) performed in patients with TSH levels > 3 μIU/mL, was associated with lower anti-Müllerian hormone (AMH) levels and lower egg yield [19]. 

### 3.4. Evaluation and Management of Hypothalamic-Pituitary Dysfunction

The ultimate goal for anovulation caused by eugonadal hypothalamic-pituitary dysfunction is to correct the underlying etiology. The majority of endocrinopathy related infertility is reversible and can be managed medically. Patients who present with menstrual dyscrasias should undergo initial evaluation with AMH, TSH and total T4, and prolactin level. If clinical signs of hyperandrogenism are apparent, 17-hydroxyprogesterone (17-OHP), dehydroepiandrosterone sulfate (DHEA-S) and total testosterone should be measured to rule out presence of ovarian or adrenal androgen secreting tumors. It is important to consider that multiple endocrinopathies may co-exist and must be treated concurrently. 

Prior to medical management, preconceptual counseling and lifestyle optimization should be emphasized, particularly weight loss and exercise. Thyroid deficiency and hyperprolactinemia, when corrected often will lead to successful ovulation and pregnancy. 

Patients suffering with PCOS commonly require ovarian suppression with combined oral contraceptive pills (OCPs), unless conception is desired. Multiple non-contraceptive benefits exist with the use of OCPs in this setting including reversing hyperandrogenism with shrinkage of exaggerated ovarian stroma, regulation of menses and reducing the risk of ovarian and endometrial cancer [20]. Those desiring pregnancy will often require ovulation induction agents. Based on multiple large randomized clinical trials, letrozole is the superior ovulation induction agent, with the recommended second line agent being clomiphene citrate (CC) [12,21]. Some of the reported benefits letrozole has compared to CC includes lower rates of multiple gestation and achieving a triple pattern proliferative endometrium required for successful implantation. Despite these reported benefits, letrozole is a relatively newer ovulation induction agent with limited long-term data compared to CC.

## 4. Ovarian Failure

Group III ovulatory disorders is defined as ovarian insufficiency or failure with a hypergonadotrophic-hypogonadic profile that affects 5% of women with ovulatory dysfunction [22]. Women possess a finite number of oocytes that gradually declines resulting in critically low levels in the mid to late 40’s leading to clinical menopause by early 50’s. Early depletion of ovarian function before the age of 40, associated with elevated FSH or low estradiol levels, is defined as premature ovarian insufficiency (POI), formerly known as premature ovarian failure. Multiple known etiologies of POI exist, including genetic, acquired or iatrogenic causes with the majority remaining idiopathic. Appendix A outlines all ovulation disorders, summarizing the various etiologies.

### 4.1. Genetic Causes

Despite a low incidence, the most common genetic form of POI remains Turner syndrome, 45 XO, with either complete or partial X chromosome deletions. These patients commonly present with primary amenorrhea due to accelerated oocyte depletion. 

The second most common cause of POI is chromosomal aberrations. Approximately 10–30% of cases appear to be familial, primarily exhibiting an x-linked inheritance pattern with varying penetrance [23,24]. Several specific gene mutations have been proven to correlate with POI. Well described is the fragile X gene premutation (FMR1) associated with CGG triple nucleotide repeat expansions (55–200) leading to POI [25]. FMR1 is an RNA-binding protein that has suppressive effects on gene expression whereby increased CGG repeats leads to overexpression of FMR1, which downregulates oocyte development genes, causing premature follicle atresia. CGG repeats greater than 200, considered a full FMR1 mutation results in complete loss of protein function, sparing ovarian function, though causing as intellectual disability in males. FMR1 premutation expansion can occur, causing maternal transmission of a full mutation to offspring. 

Additional candidate genes have been proposed to be causative, however lack sufficient evidence. New headways illustrating causes for POI includes mutations in transcription factors such as nuclear receptor subfamily five group A member 1 (NR5A1), newborn ovary homeobox (*NOBOX*) and factor in germline alpha (*FIGLA*) that are responsible for gonadal differentiation and folliculogenesis [26]. Furthermore, recent studies are shedding light on the possibility of steroid receptor and folliculogenesis growth factor mutations. Recently, copy number variants (CNVs) and microRNAs have been explored for their involvement in POI. Although additional studies are required to validate these findings, much progress is being made in identifying precise etiologies.

### 4.2. Acquired Causes

Autoimmune thyroiditis leading to hypothyroidism is the most common autoimmune disorder associated with POF. Additionally, anti-ovarian antibodies have been demonstrated targeting steroid producing cells and gonadotropin receptors. Furthermore, there are two types of autoimmune polyglandular syndromes (APS), type I and II, associated with endocrine gland and extra-glandular tissue destruction reported to cause POI. APS type I affects younger children, with 60% of girls affected experiencing primary amenorrhea and POI. Type II causes gonadal failure in 4% of affected individuals [27]. Associated autoimmune disorders linked to APS include Addison’s disease (autoimmune adrenal insufficiency) and less commonly type 1 Diabetes Mellitus, and Celiac disease [28]. The autoimmune process involves a genetic predisposition with environmental factors leading to the accumulation of dendritic cells and superabundance of lymphocytes causing tissue damage [29]. 

Common gynecologic disorders may also contribute to accelerated reproductive aging. Endometriosis affects 10% of reproductive aged women. In addition to structural pathologies, ovarian endometriosis may lead to ovarian destruction and consequently a hypoestrogenic state. A study performed by Cahill et al. described endometriosis related diminished LH levels in serum and follicular fluid impacting ovulation [30]. Inevitably, monthly fecundity decreases from 15–20% to 2–10% in a normal healthy woman affected by even mild stages of endometriosis. 

A number of environmental toxins may lead to POI. Effects of cigarette smoking have been extensively investigated. Smoking is associated with accelerated follicular atresia, displayed by significant lower AMH and earlier onset of menopause [31]. Environmental exposures also linked to ovarian failure, although still controversial and lacking powered studies, include infections such as CMV, mumps, and tuberculosis. Tuberculosis, although uncommon in developed countries, is responsible for a considerable rate of infertility cases in developing countries accounting for 5–10% of subfertility worldwide [32]. Most notably, if ovarian tuberculosis transpires (described as latent genital tuberculosis), diminished ovarian reserve and even ovarian insufficiency may ensue [33]. 

### 4.3. Iatrogenic Causes

As oncologic treatments continue to improve, and the number of cancer survivors continue to increase, so does the rate of iatrogenic ovarian insufficiency. Cancer therapy causes oocyte depletion in several ways, the most common being inhibiting growth and proliferation of cells that support oocytes. How chemotherapeutic drugs affect immature oocytes is still unclear, however commonly used toxins including alkylating agents, work by interlinking DNA causing breaks that damage the cell [34]. Additionally, anthracyclines, platinum agents as well as treatment regimens that include procarbazine [35], exhibit high rates of ovarian failure [36]. As most cancer treatment are multidrug protocols, newer studies are focused on effects of various regimens on gonadal function. Dose related effect on ovarian reserve is being further investigated. 

Pelvic and total body irradiation has significant ramifications on ovarian function. Radiation doses as low as 4–5 Gy results in permanent loss of ovarian function. Knowing these treatment modalities confer poor prognosis for post-therapy gonadal and uterine function, careful consideration can be made in planning for future fertility. The field of onco-fertility is ever growing, with fertility preservation becoming increasingly available. 

### 4.4. Age Related Ovarian Failure

Both the quantity and quality of available oocytes significantly diminishes once a woman reaches 35 years of age with exponential decline until menopause. 

### 4.5. Evaluation and Management of Ovarian Insufficiency

Ovarian insufficiency can be diagnosed by measuring AMH, estradiol, FSH levels and antral follicle count. Though spontaneous pregnancy is not impossible, it is certainly much more difficult to achieve without intervention. Patients faced with such a diagnosis should be provided reassurance and guidance to the alternate reproductive options currently available. 

Evaluation approach is dependent on medical history and physical examination. Reproductive aged women less than 35 years old with rapidly declining ovarian reserve warrant genetic and infectious disease testing. If genetic testing is abnormal, genetic counseling is prudent.

Patients with Turner syndrome may present with lack of pubertal milestone or primary amenorrhea. Mosaic Turner syndrome patients may be aware of their diagnosis prior to complete follicular atresia and can salvage their fertility by means of oocyte and/or embryo cryopreservation. With newer vitrification techniques, oocyte cryopreservation has become a reliable and available method for fertility preservation [37]. In fact, immature eggs can be matured in vitro and subsequently frozen for future use. Frozen oocytes have comparable fertilization rates to fresh oocytes [38]. These patients may benefit from preimplantation genetic testing for euploid embryo transfer to optimize pregnancy outcomes. 

Onco-fertility patients also greatly benefit from oocyte or embryo preservation prior to gonadotoxic therapy. In recent years, random start ovarian hyperstimulation in preparation for egg retrieval allows physicians to shorten treatment duration to proceed with chemoradiotherapy [39]. In a specific category of breast cancer patients, letrozole is co-administered with gonadotropins for a theoretical concern over elevated estradiol level stimulating estrogen receptor positive tumors [40]. Additionally, ovarian tissue cryopreservation may be offered to cancer patients who are prepubertal or cannot undergo ovarian stimulation with egg harvesting and freezing. Frozen ovarian tissue may be thawed followed by auto-transplanted upon completion or treated with in vitro maturation of oocytes. Previous studies have confirmed return of ovarian function and live births after transplanting the cryopreserved tissue [41,42]. If women requiring chemotherapy wish to omit cryopreservation, gonadotropin releasing hormone agonists (GnRHa) have shown some efficacy in preventing treatment induced POI [43]. 

IVF with donor oocytes currently remains the most successful treatment modality for women with POI or age related diminished ovarian reserve. Currently available options of fresh or frozen or known vs. unknown donors provides patients with more options. Furthermore, women may opt to have their young relatives, with abundant ovarian reserve, to donate oocytes to them [44]. 

The use of mesenchymal stem cells (MSCs) has been emerging as a treatment option for patients with POI. Stem cell therapy has particularly been studied in the setting of iatrogenic ovarian destruction and shown great promise. Various sources of stem cells under investigation for this purpose include umbilical cord MSCs (UCMSCs), induced pluripotent adult stem cells (iPSCs), and embryonic stem cells. Mouse models with chemotherapy induced ovarian insufficiency undergoing subsequent treatment with UCMSCs have demonstrated improved ovarian function with recovery of sex steroid levels and decreased cumulus cell apoptosis [45,46]. The suggested mechanism of ovarian rescue is explained by prevention of granulosa cell apoptosis. This maintains FSH receptor (FSHR) expressivity, and in turn prevents overproduction of FSH. As FSH exceeds a normal range secondary to low FSHR activity, accelerated follicular depletion is observed [47]. Additional studies have demonstrated administration of bone marrow MSCs restoring ovarian function in a similar manner. 

More novel therapeutic options are beginning to make strides for age related oocyte quality improvement. One suggested mechanism responsible for depleted ovarian reserve involves mitochondrial dysfunction [48]. Though still early in its experimental stage, autologous germline mitochondrial energy transfer (AUGMENT) has been performed with live births reported [49]. The utility of AUGMENT has limitations in patients with POI due to the need for abundant mature oocytes. 

Those affected by POI have health implications if they remain in a chronic hypoestrogenic state including as cardiovascular risk, sexual function and bone health. Hormone replacement therapy (HRT) should be encouraged through use of estrogen therapy with cyclic progestin if a woman still has her uterus. Those who no longer have a uterus, estrogen replacement alone is recommended. HRT is important to reduce the risk of morbidity and mortality as it mimics normal ovarian function until the natural age of menopause [50]. Dose titration and discontinuation of HRT should be considered at the average age of menopause, approximately 50 years of age.

## 5. Conclusions

In summary, this review outlines an up to date classification of ovulatory disorders to serve as a guide for its evaluation and management. As ongoing research in etiology and pathogenesis emerges, novel treatment approaches continue to develop. Familiarity with ovulatory disorders will enable practitioners to provide high quality medical care and be a beacon of hope to a wide array of patients.

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
