# Peer review of "Hypothalamic-Pituitary-Ovarian Axis Disorders Impacting Female Fertility"

_biomedicines, 2019, doi:10.3390/biomedicines7010005_

Reviewer 1 Report

The paper is well written with good literature bases assessment. There is one concern: Are you doing a plain literature reviw or are you doing a meta-analysis with concluding comments? This must be clarified in the Introduction. Are you familiar with the difference? Now you just state that you have read relevant papers. That is not enough for a scientitific assessment of the knowledge we have achieved through many studies.. Please clarifiy. beside that the information you have collected will be very valuable for our colleagues working in this field. . 

Author Response

It is now more clearly stated stated that this is in fact a review article based on recent literature. This paper is not a metanalysis. 

Reviewer 2 Report

Overall, the manuscript by Mikhaelet al. is well written. But several minor points should be concerned.

 Firstly, the structure of text is imbalanced. The introduction part is too concise. More information about causes of ovulation disorders should be mentioned in the introduction section.  

 Secondly, the title is not appropriate. Hypothalamic-pituitary-ovarian (HPO) axis should be included in the title since the authors mainly focus on HOP axis dysfunction related ovulation disorders. 

 Thirdly, the authors should combine sections 2 and 3, combine sections 4 and 5, and combine sections 6 and 7. Not sure why the authors talked about ‘evaluation and treatment’ for group I and group II disorders but focused on ‘management’ for group III disorders. They should be consistent.

 Fourthly, the authors should design a diagram to summarize the manuscript.

 Therefore, the current version of manuscript is not qualified for publication at Biomedicines.

Author Response

The introduction has the various etiologies listed as to prepare the reader for causes of each category of ovulation disorders and what will be discussed in the body of the review article

The title has been adjusted to Hypothalamic-Pituitary-Ovarian Axis Disorders impacting fertility

Although evaluation and management was discussed in group I-III disorders, you are correct the subtitles were incorrect and inconsistent. This has been corrected.

A diagram has been included in a separate file to summarize the article.

All changes have been tracked 
